# Effects of Real-Time Feedback Methods on Static Balance Training in Stroke Patients: A Randomized Controlled Trial

**DOI:** 10.3390/healthcare12070767

**Published:** 2024-04-01

**Authors:** Il-Ho Kwon, Won-Seob Shin, Kyu-Seong Choi, Myung-Sun Lee

**Affiliations:** 1Department of Physical Therapy, Graduate School of Daejeon University, 62, Daehak-ro, Dong-gu, Daejeon 34520, Republic of Korea; ilho9015@korea.kr; 2Department of Physical Therapy, College of Health Medical Science, Daejeon University, 62, Daehak-ro, Dong-gu, Daejeon 34520, Republic of Korea; 3Department of Physical Therapy, Graduate School of Health and Medicine, Daejeon University, 62, Daehak-ro, Dong-gu, Daejeon 34520, Republic of Korea; rbtjd15973@naver.com; 4Department of Beauty Design, College of Design and Art, Daejeon University, 62, Daehak-ro, Dong-gu, Daejeon 34520, Republic of Korea; leesun@dju.kr

**Keywords:** static balance, knowledge of result, knowledge of performance, stroke

## Abstract

Background: The purpose of this study was to investigate the effects of real-time feedback methods on static balance training in stroke patients. There are two types of real-time feedback methods, as follows: one is Knowledge of Result (KR), and the other is Knowledge of Performance (KP). Method: Thirty stroke patients participated in this study and were randomly assigned to the KR group (n = 15) or the KP group (n = 15). All of the groups underwent real-time feedback training for four weeks (30 min per session, five sessions per week). The primary outcomes were sway length, sway velocity, and area 95%, which were assessed before and after the intervention. The secondary outcomes included the Berg Balance Scale, the Fugl Meyer Assessment for Lower Extremity, the Postural Assessment Scale for Stroke Trunk Impairment Scale, and the Fall Efficacy Scale. A group × time interaction was assessed using two-way ANOVA with repeated measures. Result: There was a significant increase over time in all outcomes (*p* < 0.05). Significant differences were observed for a group × time interaction in sway length and area 95% (*p* < 0.05). Conclusions: Real-time feedback training for static balance enhanced stroke patients’ static balance abilities, clinical outcome assessments, and promoted self-efficacy against falls.

## 1. Introduction

A stroke is caused by a physiological or physical interruption in brain blood flow, resulting in damage to the functional brain network. It has been cited as the leading cause of physical disability in adults, leading to physical disabilities such as loss of motor skills, unilateral paralysis, and balance disorders. These disabilities make it challenging for stroke patients to perform daily activities, including walking safely, exposing them to the risk of falls [1].

The causes of falls in stroke patients include reduced muscle strength on the paralyzed side, decreased mobility, and abnormal muscle activity. Additionally, there are balance problems in static positions like sitting and standing due to asymmetry in weight bearing during static posture, reducing static balance, and hindering movement initiation [2]. In addition, functional impairment of the lower extremities is one of the most common complications in stroke patients, with reduced functional movement and muscle strengthening [1,2]. Balance and gait training have shown positive effects in preventing falls in systematic review and meta-analysis study. Training with wearable sensors has proven to be effective by providing immediate and sensitive feedback. This feedback allows real-time verification of movements and quick error correction, enhancing motor learning [3]. Motor learning, associated with neuroplasticity, aids in improving physical deficits in stroke patients [4,5]. Training with feedback promotes neuroplasticity, stimulates the brain, and facilitates motor recovery [6,7,8].

Feedback on maintaining static balance can be either intrinsic or extrinsic feedback. Intrinsic feedback can control the balance using sensory systems (e.g., visual, vestibular, proprioception, etc.) inside the body. Extrinsic feedback can control the balance using additional external information (e.g., verbal comment from a therapist or extra audio-visual information, etc.). Feedback is used as a mechanism for correcting movement errors and learning new movement skills through a process of self-control [9]. Static balance training is often used in stroke patients who have difficulty with intrinsic feedback training due to brain damage. This provides information using extrinsic feedback to learn new movement skills, commonly through balance training on a force plate with visual feedback [10]. One popular option for visual feedback-based training, specifically balance training, which provides continuous visual feedback about the center of pressure (COP) while the participant is standing on a force platform and instructs the participant to minimize movement of the COP while standing quietly, has been shown to be more effective than conventional therapy without feedback in relearning and improving balance, and improving mobility and gait in stroke patients [1,10]. There are two types of extrinsic feedback, as follows: performance outcome information and exercise performance information, which can be categorized as knowledge of result (KR) feedback or knowledge of performance (KP) feedback [11]. Although the aforementioned KR feedback and KP feedback have different roles and functions [12], this study will utilize feedback in real-time from the perspective of how to provide information about errors and promote learners to correct them rather than strengthen (or weaken) the stimulus–response relationship. Previous research on visual feedback has only been conducted using KR feedback (i.e., if the COP is right, give feedback that it is right) [13,14]. Since no studies have used KR or KP feedback in real-time, we used COP to measure static and dynamic balance and analyzed their correlation with the Berg Balance Scale to evaluate balance [15]. Therefore, in this study, we will use COP to provide real-time feedback training, and the Berg Balance Scale for balance ability, as follows. KR feedback is feedback about the position of the COP in real-time, which the subject thinks about and adjusts in order to maintain balance, and KP feedback is feedback about the COP in real-time, which the subject must perform in order to maintain balance.

The purpose of this study was to investigate the effects of two different real-time feedback methods on static balance in stroke patients. In other words, this study aims to determine the effectiveness of training stroke patients in maintaining a balanced static posture through real-time KR feedback or KP feedback on key variables of static balance maintenance and clinical evaluation. Ultimately, this study seeks to determine the effectiveness of two different methods of postural maintenance using COP, utilizing visual feedback among other methods of feedback.

## 2. Materials and Methods

### 2.1. Study Design

This study employed a single-blind, randomized controlled trial design. A two-group, pre-test-posttest design was utilized to assess and compare both primary and secondary outcomes. Evaluations were conducted by a physiotherapist with 10 years of experience before and after the intervention. The primary outcome involved kinematic data confirming static balance, including sway length (cm), sway velocity (cm/s), and area 95 (cm^2^), assessed using a force plate. Additionally, clinical assessment tools were employed to gauge the intervention’s effectiveness as a secondary outcome. The intervention spanned four weeks, occurring five times a week for 30 min each session. The study included a total of 30 participants, who were randomly assigned to either the Knowledge of Result Group (KR, n = 15) or the Knowledge of Performance Group (KP, n = 15). Three subjects were excluded during the study due to hospitalization (KR group n = 2, KP group n = 1) (Figure 1).

### 2.2. Participants

This study is for patients hospitalized at Y Hospital in Daejeon Metropolitan City. The inclusion criteria are stroke patients who can independently maintain a standing position for more than 2 min and a Berg Balance score of 21–40 (intermediate risk for falls). They were considered to have the cognitive ability to understand the study with a Mini Mental State Examination Korean (MMSE-K) score of 24 or more. Exclusion criteria are orthopedic problems, neurological problems other than stroke, and visual impairment. The purpose and method of the study were explained, and consent was obtained. This study was approved by the Ethics Committee of Daejeon University (IRB-1040647-202304-HR-005-03). G*Power software (G*Power 3.1.9.7, University of Kiel, Kiel, Germany) was used to calculate the appropriate sample size. Furthermore, with reference to the main effect size d of 1.38 reported by Karasu et al. [16], including a significance level alpha of 0.05 and a power 1-β of 0.95, the sample size was calculated to be 15 for each group.

### 2.3. Outcome Measures

#### 2.3.1. Primary Outcomes (Kinematic Data)

The primary outcome measure was static balance. A force plate (Model BP400600, AMTI, Waterton, MA, USA) was used to measure static balance ability. The force plate specifications are 400 mm × 600 mm, and it is a tool that allows for continuous measurement and recording of the subject’s COP [15]. The subject is positioned on the force plate and maintains a comfortable standing position. The subject is asked to stare at a circular dot 2 m in front of them. A total of three times, for a period of 1 min, the eyes are opened and closed. For the safety of the subject between measurements, a therapist monitored the patient’s fall from a very short distance (especially when the eyes were closed). The variables measured were sway length (cm; total length in all directions of the COP), sway velocity (cm/s; average velocity of the COP), and area 95% (cm^2^; 95% of the total area moved by the COP), which were measured before and after the intervention [17]. The sway length, sway velocity, and area 95% were subsequently calculated by the BioAnalysis Version 2.2 (AMTI, BioAnalysis, Version 2.2, Watertown, MA, USA). The amplitude of displacement reflects the distance between the maximum and minimum COP displacement for each direction, with larger values indicating poorer postural stability. The average COP velocity reflects the efficiency of the postural control system, and the smaller the velocity, the better the postural stability [18].

#### 2.3.2. Secondary Outcomes (Clinical Data)

Secondary outcome measures were measured to determine clinical differences before and after the intervention. The assessment tools were the Korean Version of the Berg Balance Scale, the Korean Version of the Fugl Meyer Assessment Lower Extremity, the Postural Assessment Scale for Stroke for the Korean, the Korean Version of the Trunk Impairment Scale, and the Korean Fall Efficacy Scale to assess confidence in falling. Since the assessment was conducted on Korean people, Korean versions of the assessment tools were used, and validity or reliability verification was completed.

The Korean Version of the Berg Balance Scale (BBS) is used to assess the balance ability in research subjects. It is a tool to predict walking ability in stroke patients, the rationale for the use of assistive devices (such as canes, wheelchairs, etc.), and the likelihood of independent walking upon return to the community. It consists of fourteen items and scores from 0–56, with 0–20 being wheelchair (high risk for falls), 21–40 being orthosis (intermediate risk for falls), and 41–56 being able to walk independently (low risk for falls). Inter-rater reliability was 0.97 and intra-rater reliability was 0.96 [19].

The Korean Version of the Fugl Meyer Assessment Lower Extremity (FMA-LE) consists of 17 lower extremity movements. Its items are scored from 0–2 and the score of the FMA-LE has a range of 0–34. It was administered in supine posture, prone posture, sitting posture, and standing posture. In the supine posture, hip flexion, hip extension, hip adduction, knee flexion, knee extension, ankle dorsiflexion, ankle plantarflexion, heel–shin speed, heel–shin tremor, and heel–shin dysmetria were assessed; in the prone posture, a hamstring reflex test and ankle plantar-flexor reflex test were conducted; in the sitting posture we assessed knee extension, ankle dorsiflexion, and conducted a knee extensor reflex test; and in the standing posture, we assessed knee flexion and ankle dorsiflexion. The reliability of the intra-class correlation coefficient was 0.96 [20].

The Postural Assessment Scale for Stroke for the Korean (PASS) is a complementary assessment tool to the FMA, with a total of twelve items, including five items for postural maintenance and seven items for postural change. It can assess postural control performance in stroke patients. PASS has been utilized as a useful clinical tool to identify stroke patients. Each item is scored from 0–36, with higher scores indicating a greater ability to maintain and change posture. In correlation with the Functional Independence Measure, the construct validity is 0.73, the inter-rater reliability is 0.88, and the test-retest reliability is 0.72 [21].

The Korean Version of the Trunk Impairment Scale (TIS) is a tool to assess static and dynamic balance, trunk movement, and coordination in stroke patients in a sitting position. It consists of seventeen items, three for static sitting balance, ten for dynamic sitting balance, and four for coordination, ranging from 0–23. The inter-rater reliability is 0.85–0.99, and the test-retest reliability is 0.96–0.99 [22].

The Korean Fall Efficacy Scale (FES) is a self-report questionnaire with ten items of daily living activities. Self-efficacy is an individual’s perception of their confidence in performing the ten items on a 10-point scale. The higher the score, the less fear of fall. The internal consistency Cronbach’s alpha of the FES is 0.90, and the test-retest reliability is 0.73 [23].

### 2.4. Interventions

Both groups underwent traditional physical therapy at the hospital, supplemented with 30 min of training involving KR and KP feedback. Traditional physical therapy was performed by a physical therapist once or twice a day for 30 min. Proprioceptive Neuromuscular Facilitation techniques, neurodevelopmental therapy, and Bobath therapy were performed twice a day for 30 min each, and pain therapy and electrotherapy were performed once each. All subjects were treated by the same physiotherapist five times per week for four weeks, and all subjects were treated by the same physiotherapist for four weeks. Throughout the feedback training, exercises were tailored to each subject’s ability, and therapists were present to prevent falls. The training was performed by a trained physiotherapist, and simple stretches were performed before and after the training (5 min each). Additionally, feedback training is the practice of keeping the center against external disturbances from the physiotherapists. The external disturbances (intermittent and continuous resistance, at a level that is challenging for the subject but safe for the therapist) were front–back, left–right, and diagonal (20 min) [2,14]. The feedback was presented to participants via a black box measuring 30 cm by 30 cm at eye level, positioned 2 m in front of them. The box featured nine LEDs for real-time feedback display. In the KR group, participants received feedback indicating the location of their COP, as illustrated in Figure 2a. Meanwhile, the KP group received feedback indicating the direction in which they should move, as depicted in Figure 2b. Both forms of feedback were presented in real-time, with the KR group receiving messages like “you missed center”, and the KP group receiving instructions such as “you need to move left or right, etc.” (Figure 2). For example, if the subject’s COP in the KR group is on the right, feedback is given that it is on the right, despite the external disturbances from the therapist. In response, the subject shifts to the left to center their own COP. On the other hand, if the KP subject’s COP is on the right, the feedback is to move to the left. In response, the subject moves their COP in response to the feedback.

Feedback was obtained from four areas under the research participants’ feet (big toe, head of the first metatarsal bone, head of the fifth metatarsal bone, and heel) to collect COP data. The four areas under the feet collect data using a force sensing resistor (FSR, a round, 0.5″ diameter) to represent the participant’s COP. These COP data were utilized for feedback-based training. 

### 2.5. Statistical Analysis

The collected data were analyzed using the IBM SPSS Statistics version 25.0 program (IBM Co., Armonk, NY, USA). The general characteristics of the subjects were expressed as mean and standard deviation using descriptive statistics. Homogeneity was confirmed using chi-squared tests and *t*-tests. Normality tests for all variables were performed using the Shapiro–Wilk test. To compare changes over time and between the two groups, a two-way ANOVA with repeated measures was conducted. The statistical significance level was set at 0.05.

## 3. Results

### 3.1. General Characteristic of the Subjects

The results of the comparison of general characteristics of the research participants showed, as presented in Table 1, that there were no statistically significant differences (Table 1).

### 3.2. Primary Outcome Measures

The primary outcome measures of feedback type showed significant differences over time for all variables. Significant differences for the group × time interaction were observed only in the eyes-open sway length and area 95% (Table 2).

### 3.3. Secondary Outcome Measures

The results of the secondary outcome measures by feedback type showed significant differences over time for all of them, but no significant differences for group × time (Table 3).

## 4. Discussion

The purpose of this study was to determine the effects of real-time knowledge of result feedback and knowledge of performance feedback training on static balance in stroke patients. As a result, significant differences were observed over time in both primary outcomes and secondary outcomes. Depending on the group, when measured with eyes open in primary outcomes, there were significant differences in sway length and area 95%.

There was no control group in this study. However, there was a previous study that compared a control group, an exercise without feedback group, and an exercise with feedback group, and they found that only the exercise feedback group was effective [24]. Other studies have compared two different training methods without a control group [14]. Both groups demonstrated significant improvements in the primary outcome measures over time. As indicated in previous studies, we attribute these improvements to the provision of real-time visual feedback on the subjects’ center of pressure, prompting the subjects to train themselves in maintaining static balance. Previous research has consistently demonstrated that static balance training enhances subjects’ static balance abilities, with visual feedback on the center of pressure contributing to improved proprioception [25]. In a study focused on real-time static balance training, visual feedback was shown to enhance spatial awareness, leading to improved postural control as a foundation for enhanced static balance ability [26]. Furthermore, effective maintenance of a stable static standing posture requires integrated input from the visual, proprioception, and motor systems [27]. In this study, we corroborated the findings of previous research, confirming that static balance abilities (sway length, sway velocity, area 95%) were significantly enhanced through real-time visual feedback training.

In the between-group comparison, the KR group exhibited significant improvement solely in static balance ability compared to the KP group. Deutsch et al. [28] discovered that the KR group received more feedback, both in terms of quality and quantity, than the KP group. Furthermore, they found that the evaluators perceived the training with KR as more valid than the training with KP. In our study, we exclusively employed real-time feedback on the center of pressure among various real-time feedback methods. It is posited that the KR group received superior feedback in terms of quality, as they engaged in stable static balance training by identifying their own center of pressure. However, as both groups were subjected to training solely using real-time feedback, there exists no quantitative difference between them.

The absence of a difference with eyes closed was also observed in a previous study utilizing real-time visual feedback training. In that study, a distinction in the center of pressure was noted with eyes open but not with eyes closed [29]. This lack of differentiation with eyes closed is likely attributed to the compromised sensory and motor functions resulting from brain damage in stroke patients. Stroke can impact various neurological functions, including sensory perception, proprioception, and motor control, which are essential components for maintaining balance. Particularly in individuals affected by stroke, sensory input and integration required for balance may be significantly impaired, leading to challenges in maintaining balance, especially in the absence of visual input (i.e., eyes closed). Consequently, evaluating static balance with eyes closed can unveil the difficulties faced by stroke patients in maintaining their static balance due to underlying neurological impairments.

The significant improvement over time in secondary outcome measures can be attributed to both groups participating in a four-week training program utilizing real-time visual feedback to enhance their static balance abilities. This approach is particularly valuable for stroke patients, as many of them experience a loss of proprioception, making it challenging to maintain proper posture. Therefore, the use of visual feedback becomes crucial for effective balance training [30]. Research indicating that visual feedback is the most effective among sensory feedback methods led to the implementation of real-time feedback training with visual feedback methods in this study. Previous studies employing real-time feedback training have emphasized its ability to activate essential variables for brain reorganization (i.e., neuroplasticity) through cognitive, functional, and complex tasks. This activation involves the prefrontal cortex, primary motor cortex, supplementary motor area, and cerebellum, rendering real-time feedback training effective [31]. Stroke patients often struggle with maintaining symmetrical posture due to reduced proprioception and weak muscle strength on the paralyzed side, leading to instability during upright posture or movement [32]. Previous research on static balance training has demonstrated improvements in fall risk, weight shifting, movements, and gait through static balance training alone. These studies have also recommended incorporating more detailed and varied types of feedback [33].

In the present study, real-time feedback training significantly improved all secondary outcomes over time. Therefore, it is believed that real-time feedback training, especially for stroke patients with impaired static balance abilities, can be effective, as it provides accurate feedback for training and enhances overall balance control. Notably, there were no significant differences in secondary outcomes between the groups in this study. While previous studies reported significant differences in clinical assessments such as BBS, FMA-LE, PASS, TIS, and FES [1,16,34], it is posited that these differences were due to comparisons with groups that did not receive any training. In this study, both groups employed training methods utilizing real-time feedback, which differed from the previous studies. The inclusion of functional assessments related to dynamic balance and movement in the evaluation is also believed to contribute to the lack of significant differences.

Significant changes in fall efficacy over time were observed in both groups, as both received real-time feedback training in addition to traditional physical therapy. Although participants were randomized for training, there was no distinction between the two groups because both were aware of the additional training. Given that both groups underwent training related to real-time feedback, both the KR and KP groups were effective in terms of secondary outcomes. The absence of a difference between the groups precludes the assertion that one group was more effective than the other.

This study has several limitations. There was no control group in this study. It was not feasible to compare the effects on specific neural pathways that distinctly differentiate between knowledge of result feedback and knowledge of performance feedback. Despite recruiting participants with a planned 10% dropout rate, the actual dropout rate exceeded 10%, and the initially intended sample size could not be achieved. 

Stroke patients often contend with a loss of sensory and motor abilities due to brain damage, resulting in diminished balance capabilities. Consequently, stroke patients face an increased risk of falls and necessitate training to enhance their balance. Given the current dearth of research on static balance training, there is a crucial need to explore various task-oriented training methods, such as standing on both feet and standing on one foot, especially in unstable environments, with the incorporation of real-time feedback. This exploration would be invaluable for informing future research endeavors.

## 5. Conclusions

This study compared the effects of receiving feedback in real time and compared knowledge of result feedback training to knowledge of performance feedback training. Stroke patients who received knowledge of result feedback training improved their static balance more than those who received knowledge of performance feedback training. This improvement can be attributed to subjects receiving real-time feedback on their knowledge of the results, prompting them to make judgmental movements to maintain balance. Therefore, we recommend postural maintenance training with KP feedback, which is a self-determined movement training, to effectively improve static balance in stroke patients during training with visual feedback.

## Figures and Tables

**Figure 1 healthcare-12-00767-f001:**
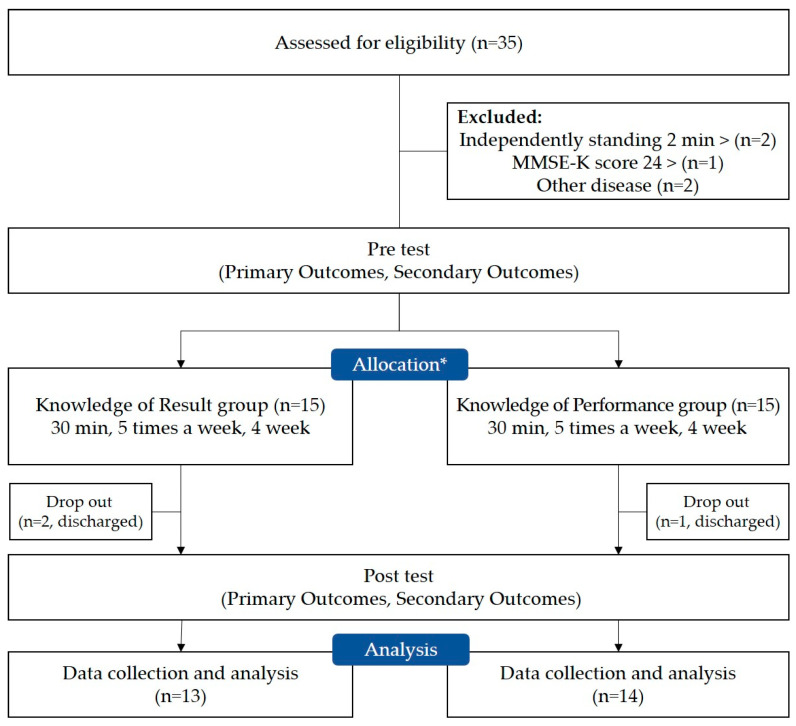
Flow chart of this study. *: randomization; MMSE-K: Mini Mental State Examination Korean.

**Figure 2 healthcare-12-00767-f002:**
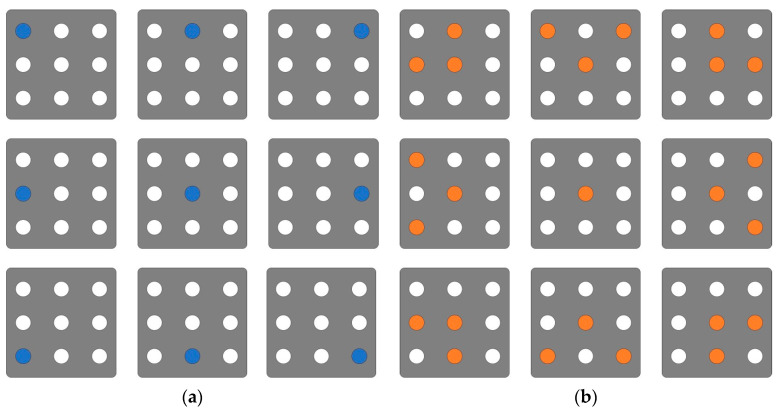
The methods of real-time feedback. (**a**) Knowledge of Result; (**b**) Knowledge of Performance.

**Table 1 healthcare-12-00767-t001:** General characteristics.

Variable	KR Group (n = 13)	KP Group (n = 14)	*p*
Sex (male/female)	9/4	9/5	0.785 ^a^
Paretic side (left/right)	7/6	8/6	0.863 ^a^
Type (infarction/hemorrhages)	9/4	5/9	0.082 ^a^
Disease duration (months)	24.92 ± 14.86	19.35 ± 15.27	0.347 ^b^
Age (years)	64.53 ± 12.35	63.14 ± 12.08	0.769 ^b^
Height (cm)	168.62 ± 8.84	164.29 ± 8.88	0.217 ^b^
Body mass (kg)	68.69 ± 7.88	65.86 ± 10.95	0.451 ^b^

All values are shown as Mean ± Standard deviation; KR: Knowledge of Results; KP: Knowledge of Performance; ^a^ Chi-square test between two groups; ^b^ independent t-test between two groups.

**Table 2 healthcare-12-00767-t002:** Within- and between-group comparisons for primary outcome measures.

Variable	State	Group	Pre	Post	Time	Group × Time
					F (*p*)	F (*p*)
Sway Length (cm)	EO	KR	387.28 ± 91.62	320.33 ± 97.91	54.614 (0.000) *	5.398 (0.029) *
		KP	402.71 ± 45.86	367.78 ± 75.87		
	EC	KR	433.82 ± 113.66	395.60 ± 112.04	52.323 (0.000) *	1.444 (0.241)
		KP	443.66 ± 81.42	410.92 ± 80.26		
Sway Velocity (cm/s)	EO	KR	6.35 ± 0.54	5.97 ± 0.41	40.735 (0.000) *	0.108 (0.745)
		KP	6.73 ± 0.62	6.33 ± 0.50		
	EC	KR	7.25 ± 0.64	6.90 ± 0.63	57.357 (0.000) *	0.065 (0.801)
		KP	6.76 ± 0.74	6.44 ± 0.63		
Area 95% (cm^2^)	EO	KR	4.10 ± 1.09	2.92 ± 1.14	113.023 (0.000) *	7.106 (0.013) *
		KP	3.38 ± 1.15	2.67 ± 1.11		
	EC	KR	6.51 ± 1.63	5.28 ± 1.65	62.656 (0.000) *	0.589 (0.450)
		KP	6.70 ± 2.79	5.69 ± 2.44		

All values are presented as Mean ± Standard deviation; KR: Knowledge of Results; KP: Knowledge of Performance; EO: Eye Open; EC: Eye Closed; * *p* < 0.05.

**Table 3 healthcare-12-00767-t003:** Within- and between-group comparisons for secondary outcome measures.

Variable	Group	Pre	Post	Time	Group × Time
				F (*p*)	F (*p*)
BBS (score)	KR	31.84 ± 9.40	38.07 ± 10.59	83.037 (0.000) *	0.559 (0.462)
	KP	32.14 ± 10.60	37.42 ± 10.29		
FMA-LE (score)	KR	19.23 ± 7.81	29.92 ± 10.51	37.101 (0.000) *	4.100 (0.054)
	KP	27.50 ± 8.02	32.85 ± 7.99		
PASS (score)	KR	26.23 ± 7.40	29.46 ± 6.23	27.821 (0.000) *	1.772 (0.195)
	KP	28.00 ± 6.73	29.92 ± 6.39		
TIS (score)	KR	15.76 ± 2.65	18.92 ± 2.59	62.510 (0.000) *	0.152 (0.700)
	KP	16.21 ± 4.04	19.07 ± 3.07		
FES (score)	KR	58.00 ± 22.20	63.84 ± 21.10	79.329 (0.000) *	0.575 (0.455)
	KP	63.00 ± 19.93	67.92 ± 20.95		

All values are showed Mean ± Standard deviation; BBS: Berg Balance Scale; FMA-LE: Fugl Meyer Assessment Lower Extremity; PASS: Postural Assessment Scale for Stroke; TIS: Trunk Impairment Scale; FES: Fall Efficacy Scale; KR: Knowledge of Results; KP: Knowledge of Performance; * *p* < 0.05.

## Data Availability

Data are contained within the article.

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
