# Peer review of "Effects of Real-Time Feedback Methods on Static Balance Training in Stroke Patients: A Randomized Controlled Trial"

_healthcare, 2024, doi:10.3390/healthcare12070767_

Round 1

Reviewer 1 Report (Previous Reviewer 1)

Comments and Suggestions for Authors

Please correct the items below carefully.

1)     In the background, please state that the purpose of this study is to compare the effectiveness of two real-time feedback methods.

Example) The purpose of this study was to investigate the effects of two different real-time feedback methods on static balance in stroke patients.

2)     In Line 188, you missing ‘%’ – area 95%

3)     Line 270, The name of the program used to calculate sway length, sway velocity, etc. does not appear to be listed.

4)     In Intervention, although traditional physical therapy was administered to both groups, it is necessary to describe what this physical therapy consisted of.

5)     Please provide the full name of the abbreviations used in the table through comments.

6)     In Figure 1, both groups are labeled as knowledge of result group. Also, the pictures are duplicates. Please fix.

Author Response

Reviewer 2 Report (Previous Reviewer 2)

Comments and Suggestions for Authors

RESULTS

My comments:

Table 1 – “Body mass” instead “Weight”…

To be correct from a scientific point of view, the unit of weight is N (Newton) and not kg…

Your response:

I know what you're talking about, In most healthcare or related medical articles, a person's weight is expressed as “Weight”. So here are a few examples that most studies use kg rather than N to represent human weight; we attach a table of previous studies.

My response: Because it is in previous articles, it does not mean that it is correct... In fact, it is not correct... For some reason there is the parameter body mass index... it is not weight index…

Author Response

This manuscript is a resubmission of an earlier submission. The following is a list of the peer review reports and author responses from that submission.

Round 1

Reviewer 1 Report

Comments and Suggestions for Authors

It appears that the English text needs to be proofread throughout the paper to convey clear meaning.

ex) Line 177: The statistical significance level a was set at 0.05.

The purpose of this study was to compare the effects of training to maintain a balanced static balance using real-time knowledge of results (KR) feedback and knowledge of performance (KP) feedback.

Please provide a reference to the settings used when determining the number of participants using gpower.

When using an abbreviation, please name the abbreviation before using it.

Example) Describe the method or program used when randomizing COP subjects.

Please name the abbreviations used in the table through comments.

Please indicate the number of people who were dropped-out in figure 1.

Comments on the Quality of English Language

It appears that the English text needs to be proofread throughout the paper to convey clear meaning.

ex) Line 177: The statistical significance level a was set at 0.05.

The purpose of this study was to compare the effects of training to maintain a balanced static balance using real-time knowledge of results (KR) feedback and knowledge of performance (KP) feedback.

Reviewer 2 Report

Comments and Suggestions for Authors

Thank you for your manuscript. Your topic is interesting. However, the article has serious flaws and additional experiments are needed. Furthermore, throughout the text the terms “balance” and “postural stability” are used randomly. First, I suggest using only one of these terms throughout the text. Second, I suggest it is “postural stability”. Third, I recommend its definition in the Introduction section. Please see my comments below…

INTRODUCTION

P1L39. “…and abnormal muscle movements.” – Muscles have no movement...

P1L42. “Balance and gait training have shown positive effects in preventing falls in several studies.” – References?

3rd paragraph – Please give examples of intrinsic and extrinsic feedbacks to a better understand of the text… The text is not clear in some passages… There are acronyms that were already described previously in the text, and others that appear without any reference, i.e., COP...

P2L56. “There are two types of feedback…” – Are you referring to intrinsic or extrinsic feedback?

As previously mentioned, the need to define postural stability is clear to me; and explain why the center of pressure (COP) appears as a key feedback parameter regarding postural stability... In this way, there is a lack of explanation of how postural stability can be assessed, especially in stroke patients – this will help to understand some choices made in this study regarding the assessment of postural stability; it is also worth indicating what the gold standard parameter is...

This study aims to determine the effectiveness of training stroke patients in maintaining a balanced static posture through real-time knowledge of result (KR) feedback or knowledge of performance (KP) feedback on key variables of static balance maintenance and clinical evaluation. The Introduction section does not make clear the importance of this study in relation to this issue. Is this the first study on the topic? What's new?

MATERIAL AND METHODS

This study seeks to provide evidence of the effectiveness of intervention through feedback training. Can the study design used achieve this? In my opinion, I don't think so... It is missing a group that has done the same exercises but has not received any feedback... Only then can we understand if the improvements are the result of the feedback and not the exercises... Or in other words, understand the role of feedback...

P2L77. “The primary outcome involved kinematic data confirming static balance, including sway length (cm), sway speed (cm/s), and area 95 (cm²), assessed using a force plate.” – Sway length and speed of what? What is area 95? An explanation is needed...

Lack of information regarding inclusion and exclusion criteria: 1) according to what is defined, we can have stroke patients in which postural stability has not been affected; 2) have patients already had physical therapy before? Or was this intervention at the beginning of their recovery processes?

P3L102. “The force plate measures 400 mm600 mm...” – Model?

P3L108. “The variables measured were sway length (cm), sway speed (cm/s)…” – In which direction? Anteroposterior? Mediolateral? An explanation is needed....

P3L119. “The Korean Version of Berg Balance Scale (BBS) is used to assess the balance ability and risk of falls in research subjects.” – I have concerns about its ability to assess fall risk...

P3L126. “Korean Version of the Fugl Meyer Assessment Lower Extremity (FMA-LE) consists of 17 lower extremity movements. Its item is scored from 0 to 2 and the score of the FMA-LE has a range of 0-34. It was administered in supine posture, prone posture, sitting posture, and standing posture. In the supine posture was assessed hip flexion, hip extension, hip adduction, knee flexion, knee extension, ankle dorsiflexion, ankle plantarflexion, heel–shin speed, heel–shin tremor, and heel–shin dysmetria, in the prone posture was assessed hamstring reflex test and ankle plantar-flexor reflex test, in the sitting posture was assessed knee extension, ankle dorsiflexion, and knee extensor reflex test, and in the standing posture was assessed knee flexion and ankle dorsiflexion. The reliability of intraclass correlation coefficient was 0.96 [16].” – How important is this tool for this study, which aims to evaluate postural stability?

P3L126. “It can assess postural control performance as well as balance in stroke patients.” – What is the difference between postural control and balance?

Regarding intervention, examples of the proposed exercises are needed... Furthermore, the number of exercises, sets, and repetitions must also be described... It is very unclear... It is also unclear that a force platform or pressure plate was used...

RESULTS

Table 1 – “Body mass” instead “Weight”…

DISCUSSION

P7L214. “As indicated in previous studies, we attribute these improvements to the provision of real-time visual feedback on the subjects' center of pressure, prompting the subjects to train themselves in maintaining static balance.” – How can you make this statement? Your study does not have a group that has done the same exercises but has not received any feedback... Improvements may be the result of the exercises performed…

CONCLUSIONS

Due to the study design used, this conclusion is not supported by the results...

Round 2

Reviewer 2 Report

Comments and Suggestions for Authors

Your response:

Thank you for your valuable feedback. Upon reviewing previous research papers, it is clear that feedback training has a significant impact. The objective of this study is to investigate which of two real-time feedback training methods is more effective. While including a pure control group without feedback would have been ideal to better demonstrate the effect of feedback, unfortunately, we did not include a control group in this study. While it was possible to provide visual feedback differently as KP and KR, it was not possible to proceed without providing visual feedback. If stroke patients were left to stand still without providing feedback for the sake of a pure control group, it would have raised ethical concerns. However, for the purpose of this study, comparing the two intervention groups was deemed sufficient as the feedback-free control group was not deemed essential to achieve the study's objectives. We will take this into consideration for future research.

My comment: Your answer is clear but the Introduction section must present the same clarity as this answer... However, for me, the lack of a control group is a major limitation to this study...

P2L77. “The variables measured were sway length (cm; total length in all direction of the COP), sway speed (cm/s; average speed of the COP), and area 95% (cm2; 95% of the total area moved by the COP)…” – How was velocity calculated? The best term is velocity…

Lack of information regarding inclusion and exclusion criteria: 1) according to what is defined, we can have stroke patients in which postural stability has not been affected; 2) have patients already had physical therapy before? Or was this intervention at the beginning of their recovery processes?

Your response:

Thank you for the valuable insight. We believe that, as a stroke patient, there is generally a decreased postural stability when compared to non-disabled individuals. Moreover, since the study focused on hospitalized patients, it is likely that it included only stroke patients with compromised postural stability. Participants were recruited without specifying a recovery stage, and all subjects were enrolled during their hospitalization. Therefore, the registration and data collection took place in the early or mid-stages of hospitalization. While most subjects were in the chronic stage, two individuals were in the sub-acute stage.

My comment: For me, your answer is not satisfactory. Including subjects with an indeterminate level of affectation may have created a heterogeneous group. A possible solution could have been to define the following inclusion criteria: subjects being below a certain value on the Berg Balance Scale...

P3L126. “Korean Version of the Fugl Meyer Assessment Lower Extremity (FMA-LE) consists of 17 lower extremity movements. Its item is scored from 0 to 2 and the score of the FMA-LE has a range of 0-34. It was administered in supine posture, prone posture, sitting posture, and standing posture. In the supine posture was assessed hip flexion, hip extension, hip adduction, knee flexion, knee extension, ankle dorsiflexion, ankle plantarflexion, heel–shin speed, heel–shin tremor, and heel–shin dysmetria, in the prone posture was assessed hamstring reflex test and ankle plantar-flexor reflex test, in the sitting posture was assessed knee extension, ankle dorsiflexion, and knee extensor reflex test, and in the standing posture was assessed knee flexion and ankle dorsiflexion. The reliability of intraclass correlation coefficient was 0.96 [16].” – How important is this tool for this study, which aims to evaluate postural stability?

Your response:

Thank you for your question. As this study design was a 4-week intervention, we wanted to see changes in motor function as well as postural stability.

My comment: Nothing is mentioned about this secondary objective in the Introduction section...

Regarding intervention, examples of the proposed exercises are needed... Furthermore, the number of exercises, sets, and repetitions must also be described... It is very unclear... It is also unclear that a force platform or pressure plate was used...

Your response:

Thanks for the nice comment. I think I missed something during the editing. The intervention does not use force plates, and the COP is checked using sensors under the feet to provide visual feedback as shown in Figure 2. Therefore, we added the following. “The training was performed by a trained physiotherapist, and simple stretches were performed before and after the training (5 minutes each). And feedback training is the practice of keeping the center against external disturbance from the physiotherapists. The external disturbances were front-back, left-right, and diagonal (20 minutes). The four areas under the feet collect data using force sensing resistor (FSR; a round, 0.5" diameter) to represent the participant's COP.

My comment: However, there is information that is still missing, e.g., number of sets and repetitions...

RESULTS

Table 1 – “Body mass” instead “Weight”…

Your response:

In this paper, we're measuring by weight, so I think it's appropriate to use “weight.”

My comment: To be correct from a scientific point of view, the unit of weight is N (Newton) and not kg…
